# MAIT cell clonal expansion and TCR repertoire shaping in human volunteers challenged with *Salmonella* Paratyphi A

Lauren J. Howson [1], Giorgio Napolitani[1], Dawn Shepherd[1,5], Hemza Ghadbane[1,6], Prathiba Kurupati[1], Lorena Preciado-Llanes[1], Margarida Rei[1], Hazel C. Dobinson[2], Malick M. Gibani[2], Karen Wei Weng Teng[3], Evan W. Newell [3], Natacha Veerapen[4], Gurdyal S. Besra [4], Andrew J. Pollard[2] & Vincenzo Cerundolo[1]

Mucosal-associated invariant T (MAIT) cells are innate-like T cells that can detect bacteria-derived metabolites presented on MR1. Here we show, using a controlled infection of humans with live *Salmonella enterica* serovar Paratyphi A, that MAIT cells are activated during infection, an effect maintained even after antibiotic treatment. At the peak of infection MAIT cell T-cell receptor (TCR)β clonotypes that are over-represented prior to infection transiently contract. Select MAIT cell TCRβ clonotypes that expand after infection have stronger TCR-dependent activation than do contracted clonotypes. Our results demonstrate that host exposure to antigen may drive clonal expansion of MAIT cells with increased functional avidity, suggesting a role for specific vaccination strategies to increase the frequency and potency of MAIT cells to optimize effector function.

[1] Medical Research Council (MRC) Human Immunology Unit, Weatherall Institute of Molecular Medicine, University of Oxford, Oxford OX3 9DS, UK. [2] Oxford Vaccine Group, Department of Paediatrics, University of Oxford and the National Institute for Health Research (NIHR) Oxford Biomedical Research Centre, Oxford OX3 9DU, UK. [3] Agency for Science, Technology and Research (A*STAR), Singapore Immunology Network (SIgN), Singapore 138648, Singapore. [4] School of Biosciences, University of Birmingham, Edgbaston B15 2TT, UK. [5]Present address: Department of Pharmacology, University of Oxford, Mansfield Rd, Oxford OX1 3QT, UK. [6]Present address: Immunocore Ltd, 101 Park Drive, Milton Park, Abingdon OX14 4RY, UK. Correspondence and requests for materials should be addressed to V.C. (email: vincenzo.cerundolo@imm.ox.ac.uk)

Mucosal-associated invariant T (MAIT) cells are non-classical T cells with both innate and adaptive immune properties[1]. These cells respond to a recently identified class of antigens, vitamin B metabolites derived from bacteria, that are presented on the MHC class I-like molecule MR1[2]. This discovery, along with an abundance in humans[3,4], conservation across species[5–7], and the defective immune response of MR1/MAIT cell-deficient mice to infection by *Francisella tularensis*[8], *Klebsiella pneumoniae*[9], and *Mycobacterium tuberculosis*[10], has led to the notion that these cells have an important function in controlling bacterial infections.

MAIT cells recognize vitamin metabolites due to their T-cell receptor (TCR) usage, which in humans is an invariant TCRα chain TRAV1-2–TRAJ33 (or TRAJ12/TRAJ20 at lower frequencies)[11]. However, their TCRβ chain usage is oligoclonal, with predominantly TRBV20 and TRBV6 usage[12]. Such varied TCRβ chain usage has been suggested to enable differential antigen, and potentially pathogen, recognition[13–15].

Most studies of MAIT cells in bacterial infection have been either in vitro[10,16,17] or in vivo mouse models[10,8,18]. These studies have demonstrated that MAIT cells respond to bacteria through upregulation of activation markers and proinflammatory cytokine

**Fig. 1** MAIT cell frequency and phenotype during *S.* Paratyphi A infection. **a** Outline of the blood sampling protocol for the controlled human *S.* Paratyphi A infection. PBMCs were isolated at various time points from 25 individuals challenged with live *S.* Paratyphi A and analyzed by flow cytometry. Graphs showing the frequency of MAIT cells in the CD3$^+$ T-cell compartment represented as fold change from baseline in individuals **b** not diagnosed (*n* = 8) or **c** diagnosed with enteric fever (*n* = 17). Graphs showing the percentage of activated MAIT cells in challenged individuals **d** not diagnosed and **e** diagnosed with enteric fever analyzed by expression of CD38. Each symbol represents a participant and a key for diagnosed individual identification is provided. For diagnosed individuals, infection was diagnosed between day 4 and day 7, with black symbols representing samples collected prior to diagnosis and red symbols representing samples taken on/after diagnosis. Line indicates mean and error bars represent standard error of the mean (SEM). Statistical significance was calculated using one-way analysis of variance (ANOVA) with Dunnett's test, where *P < 0.05 and ***P < 0.001. PT, *S.* Paratyphi A

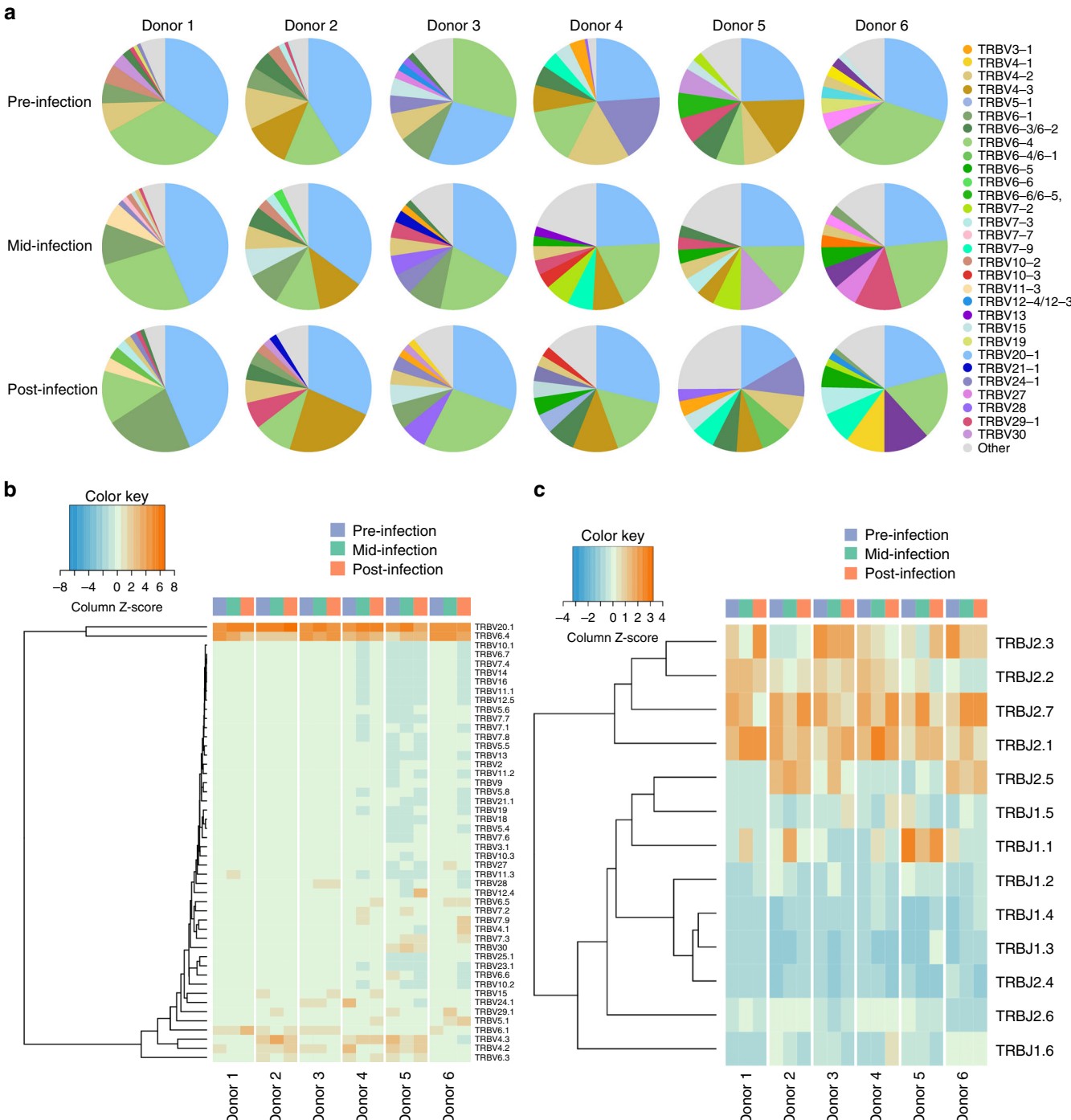

**Fig. 2** MAIT cell TCRβ usage is oligoclonal during enteric fever. MAIT cells were sorted from frozen PBMCs from six individuals challenged with *S.* Paratyphi A and diagnosed with enteric fever (pre-, mid-, and post-infection) and the TCRβ usage analyzed. **a** Pie charts of TRBV usage from diagnosed individuals. The weighted **b** TRBV and **c** TRBJ usage profile for diagnosed individuals shown as a heatmap with hierarchical clustering performed using Euclidean distance. Lower *x*-axis labels indicate diagnosed individual donors 1–6. Top *x*-axis color bars indicate sample time point

production, and only in response to bacteria that contain the riboflavin synthesis pathway[16,18], which provide a source of stimulatory vitamin B metabolites. These findings are important, as MAIT cells can also become activated in an MR1-independent manner through cytokine stimulation[19,20]. Collectively, these studies suggest that the presence of the ligand is important for MAIT cell response to bacterial infection.

Most of the few human studies of MAIT cells are cross-sectional, requiring comparison to age-matched healthy controls[16,21,22]. The only longitudinal studies of MAIT cells in response to bacterial infection involve the challenge of human volunteers with live attenuated Shigella[17], live *Salmonella enterica* serovar Typhi[23], and infection of non-human primates with *M. tuberculosis*[24]. Although these studies describe changes in the frequency and activation of MAIT cells, the TCR repertoire stability and clonal response of MAIT cells is yet to be explored.

To address this deficit, here we utilize samples from a controlled human infection model of live *S. enterica* serovar

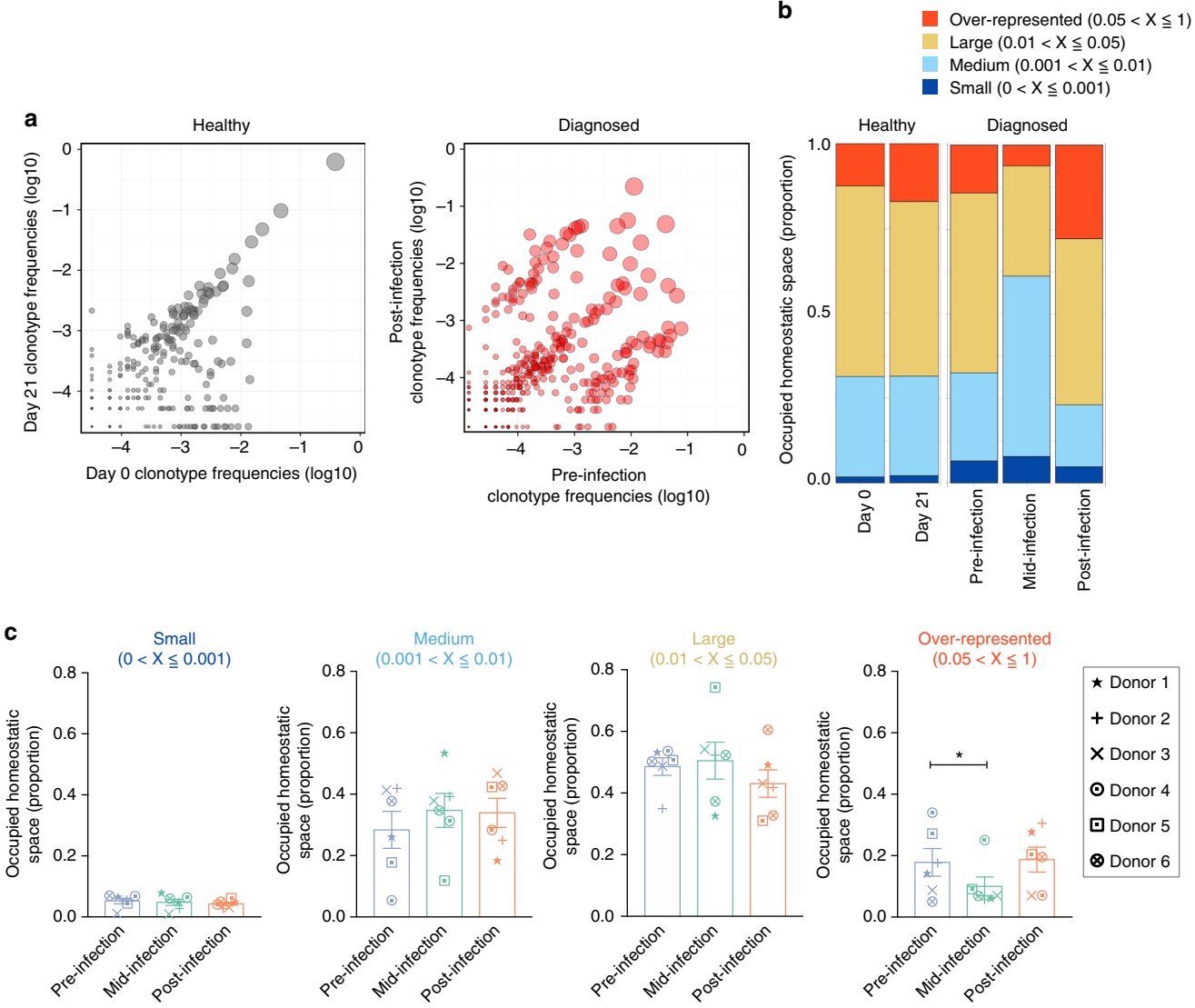

**Fig. 3** The over-represented MAIT cell clonotypes undergo transient contraction during enteric fever. MAIT cells were sorted from frozen PBMCs from six individuals challenged with *S.* Paratyphi A and diagnosed with enteric fever (pre-, mid-, and post-infection) and the TCRβ repertoire analyzed. MAIT cells from healthy individuals sampled twice (21 days apart) were analyzed as a control. **a** Examples of clonotype abundance scatterplots for a healthy and a diagnosed (donor 1) individual. Scatterplot axes represent log10 clonotype frequencies in each sample (overlapping clonotypes only) and point size is scaled to the geometric mean of the clonotype frequency in both sample time points. **b** Example graphs of the occupied homeostatic space of MAIT cell clonotypes in a healthy and diagnosed (donor 1) individual. Measured as a proportion taken up by over-represented (0.05–1), large (0.01–0.05), medium (0.001–0.01), and small (0–0.001) clonotypes. **c** Summary graphs of occupied homeostatic space taken up by MAIT clonotypes in diagnosed donor 1–6. Line is the mean with error bars representing SEM. Each symbol represents one individual and a key is provided for identification. Statistical significance was calculated using Student's paired two-tailed *t*-test, where *$P < 0.05$

Paratyphi A[25], which allows us to study the dynamic and clonal response of MAIT cells during a human invasive bacterial infection. We show that the frequency of MAIT cells decreases prior to the diagnosis of enteric fever. MAIT cells in these diagnosed individuals become activated at the peak of infection and maintain this activation, even with antibiotic treatment. TCR analysis reveals that the MAIT cell TCRβ repertoire undergoes reshaping with evidence of clonal expansion and contraction. Assessment of the specific TCRβ clonotypes suggests that one of the factors modulating the expansion of select clonotypes could be the higher functional avidity of their TCRβ to the bacterial metabolite antigen. Our results enhance our understanding of the MAIT cell clonal response and TCR repertoire during human bacterial infections.

## Results

**MAIT cells decrease in frequency during enteric fever**. We analyzed the relative frequency of circulating Vα7.2+ CD161+ MAIT cells over time in 25 healthy individuals challenged with live *S.* Paratyphi A. The MAIT cell flow cytometry/cell sorting gating strategy is outlined in Supplementary Fig. 1. Staining with MR1 tetramers[26] has provided conclusive evidence confirming the identity of circulating Vα7.2+ CD161+ MAIT cells, as shown in Supplementary Fig. 2. An overview of the human challenge trial and blood sampling protocol is shown in Fig. 1a. The frequency of MAIT cells was relatively stable in individuals not diagnosed with infection (Fig. 1b), with no significant differences ($P > 0.05$) measured by one-way analysis of variance (ANOVA) with Dunnett's test. By contrast, there was an early decrease in the

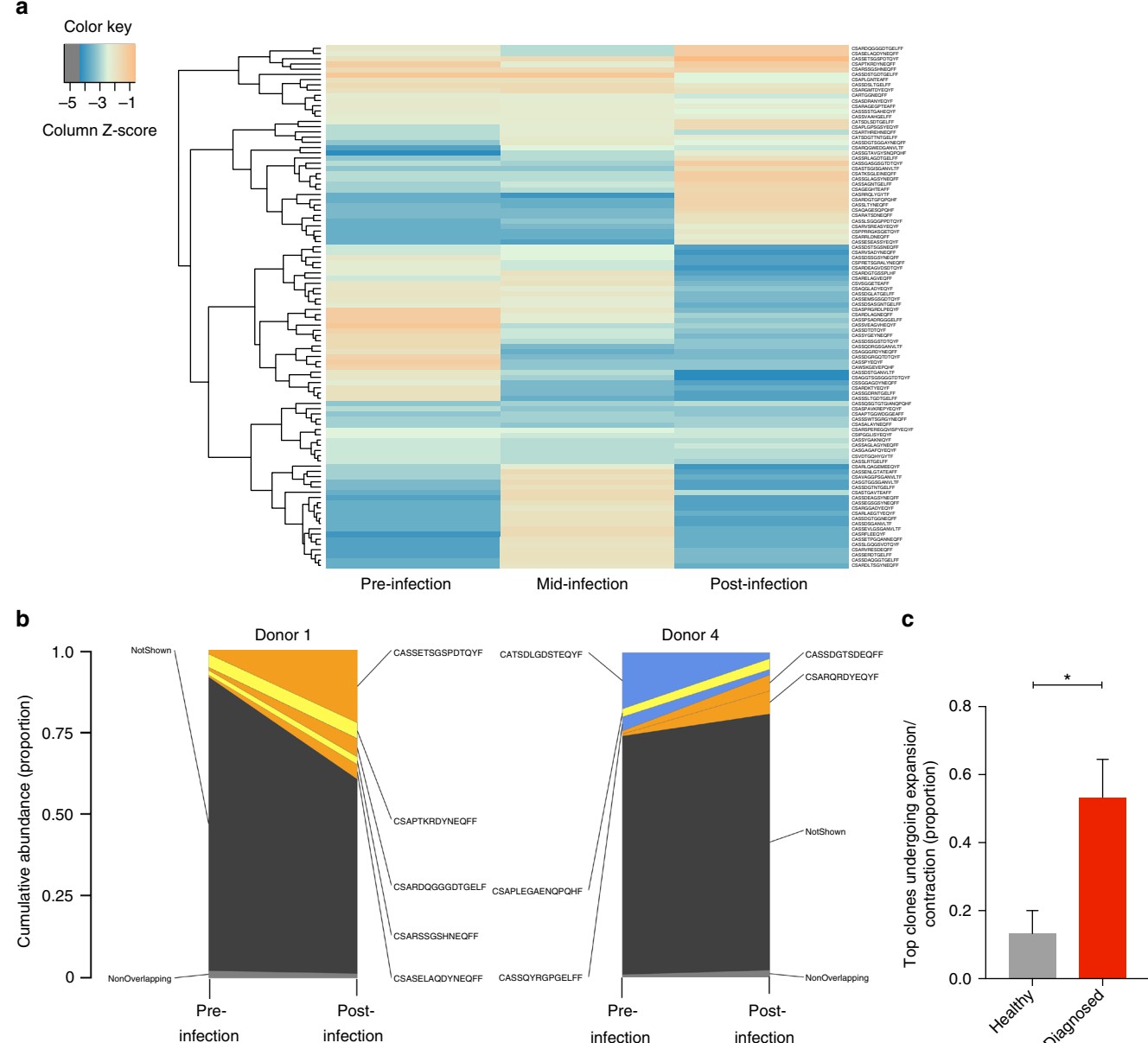

**Fig. 4** MAIT clonotype composition is altered during enteric fever. MAIT cells were sorted from frozen PBMCs from individuals challenged with *S.* Paratyphi A and diagnosed with enteric fever (pre-, mid-, and post-infection) and the TCRβ repertoire analyzed. **a** Representative clonotype tracking heatmap for a diagnosed individual (donor 1) showing top 100 clonotypes with hierarchical clustering on the left *y*-axis and the CDR3β sequences on the right *y*-axis. **b** Example shared clonotype abundance plots for diagnosed donor 1 and donor 4. The top five clonotypes shared pre- and post-infection are displayed. Orange indicates expanded clone (≥2-fold increase), blue indicates contracted clone (≥2-fold decrease) and yellow indicates a stable clone. **c** Graph summarizing the clonotype abundance plots, showing the proportion of the top five clones undergoing either contraction/expansion over time in healthy controls (*n* = 3) and diagnosed individuals (*n* = 6). Bars plotted as mean with error bars representing SEM. Statistical significance calculated using Student's unpaired two-tailed *t*-test where *P < 0.05

frequency of MAIT cells prior to diagnosis of enteric fever to 0.88 ± 0.04 on day 4, which decreased significantly to 0.63 ± 0.06 7–8 days after challenge (Fig. 1c). The frequency of MAIT cells in the diagnosed individuals then recovered after antibiotic treatment (day 28) and, in the majority (59%) of challenged individuals, we observed an increase in frequency from baseline.

**MAIT cells are activated during enteric fever.** We next examined the activation of circulating MAIT cells in challenged individuals by expression of CD38. There was no increase in the expression of CD38 over time on MAIT cells from individuals

who were not diagnosed with enteric fever (Fig. 1d). In contrast, the expression of CD38 was significantly increased on 16.8 ± 4.2% of MAIT cells 9–11 days after challenge in individuals that developed enteric fever (Fig. 1e). This significant upregulation of CD38 was maintained, even following antibiotic treatment where 16.3 ± 2.5% of MAIT cells were still expressing CD38.

To further characterize the MAIT cell phenotype during infection, mass cytometry analysis was performed on blood samples from four diagnosed individuals. We found that MAIT cells expressed the intracellular proliferation marker Ki67 following diagnosis (Supplementary Fig. 3), similar to observations made in previous bacterial and viral infection

**Table 1 TCRβ clonotypes used for Jurkat.MAIT production**

| Donor | TRBV | TRBD | TRBJ | Clone CDR3β sequence | Frequency pre-infection | Frequency post-infection | Classification |
|-------|------|------|------|----------------------|-------------------------|--------------------------|----------------|
| Donor 1 | 6-1 | 2 | 2–3 | CASSETSGSPDTQYF | 1.21% | 21.1% | Expanded |
| | 20-1 | 1/2 | 2–7 | CSASPRGRDLPEQYF | 3.4% | 0.02% | Contracted |
| Donor 3 | 6-4 | 2 | 2–1 | CASSDGTSDEQFF | 0.01% | 1.3% | Expanded |
| | 20-1 | 2 | 2–7 | CSARQRDYEQYF | 1.8% | 0.01% | Contracted |

studies[23,27]. This increased expression at the peak of infection was confirmed by flow cytometry on six additional diagnosed individuals, showing that $18.6 \pm 6.6\%$ of activated ($CD38^+$) MAIT cells expressed Ki67 at the peak of infection compared to $5.5 \pm 2.0\%$ of non-activated ($CD38^-$) MAIT cells (Supplementary Fig. 4). Interestingly, very few $Ki67^+$ cells were observed following antibiotic treatment, although $27.7 \pm 8.0\%$ of MAIT cells still retained CD38 expression (Supplementary Fig. 4). Additional analysis of surface markers revealed no substantial differences in the expression of additional activation markers or tissue-homing markers between $CD38^+Ki67^+$ and $CD38^-Ki67^-$ cells at the peak of infection (Supplementary Fig. 3).

Together, the decreased frequency and acquisition of an activated and proliferating phenotype suggests that circulating MAIT cells in individuals diagnosed with enteric fever are responding to the presence of S. Paratyphi A.

**MAIT cell TCRβ usage remains oligoclonal during enteric fever**. In vitro experiments to study the activation of MAIT cells isolated from healthy volunteers indicated that activation was partially antigen-dependent as addition of MR1 blocking antibody could partially reduce the MAIT cell response to S. Paratyphi A infection (Supplementary Fig. 5). The MAIT cell activation response to S. Paratyphi A was also partially IL-12-dependent, and a combination of both MR1 and IL-12p40 blocking antibodies prevented MAIT cell activation (Supplementary Fig. 5). This suggested that MAIT cell in vitro response was not only driven by antigen recognition, but also by cytokine-dependent activation.

We analyzed the TCRβ repertoire of six individuals challenged with S. Paratyphi A and diagnosed with enteric fever to determine whether the MAIT cell response in vivo was TCR-dependent, and therefore potentially antigen-driven. Consistent with previous findings[11,12], we found the TRBV usage of MAIT cells was oligoclonal, characterized by the preferential use of TRBV20 and TRBV6 in all diagnosed individuals, with inter-donor variation ranging from 31 to 84% of the total chain usage (Fig. 2a, b). This oligoclonality was maintained during infection and was also evident to a lesser extent in the TRBJ usage (Fig. 2c). This is similar to the TCRβ chain usage observed in healthy (not challenged, control) individuals over time (Supplementary Fig. 6). These results indicate that the TCRβ usage by MAIT cells is not influenced by the presence of S. Paratyphi A and that if changes were occurring in MAIT TCRβ repertoire that it would be on the clonotypic level.

**Over-represented MAIT cell clonotypes contract mid-infection**. To determine whether changes were occurring in MAIT cells at the clonotypic level, we analyzed circulating MAIT cell clonotypes based on the CDR3β sequences in six diagnosed individuals. In keeping with the findings of Lepore et al.[12], we confirmed that the MAIT cell clonotypes were stable over time, shown by the example clonotype scatterplot of a healthy control individual (Fig. 3a). However, MAIT cells in the diagnosed individual pre- and post-infection scatterplot appear less stable over time.

To explore this further, we analyzed the clonal space home-ostasis of MAIT cell clonotypes over time to understand whether the distribution of the clonotypes underwent changes in response to infection. In a healthy control individual, the clonal distribution was relatively stable over time. In contrast, when we analyzed the clonal distribution in a diagnosed individual (donor 1), we observed marked changes, particularly in the proportion of space used by the over-represented MAIT cell clonotypes (Fig. 3b). When comparing the clonal distribution in all diagnosed individuals, there was a significant decrease in the proportion of over-represented clones from $0.17 \pm 0.05$ at baseline to $0.10 \pm 0.04$ mid-infection, but there was no significant difference between the baseline and post-infection (Fig. 3c). There were also no significant changes to the MAIT cell clonal distribution in the undiagnosed individuals (Supplementary Fig. 7). These results demonstrate that changes to MAIT cell clonotypes in response to infection occur most notably to those that are over-represented.

**MAIT cells clonotype composition alters during enteric fever**. To determine whether the over-represented MAIT cell clonotypes following infection were different to those present before infection, we analyzed the circulating MAIT cell clonotype composition based on the CDR3β sequence in six diagnosed individuals. An example clonotype tracking heatmap for a diagnosed individual (donor 1) shows that the clonotypes present at high frequency following infection were not the same clonotypes that were at high frequency before infection (Fig. 4a). A shared clonotype abundance plot with the top five clonotypes displayed for two diagnosed individuals shows clonal expansion and contraction occurring, with a predominating clonotype present post-infection in individual donor 1 (Fig. 4b). Shared clonotype abundance plots for all diagnosed individuals are shown in Supplementary Fig. 8. When comparing the top five clones across all diagnosed individuals, we observed significantly greater proportion of top clonotypes undergoing clonal expansion/contraction ($0.53 \pm 0.11$) compared to healthy controls ($0.13 \pm 0.07$) (Fig. 4c). These results demonstrate that the composition of MAIT cell clonotypes in circulation changes in response to S. Paratyphi A infection.

**Expanded MAIT cell TCRβ clonotypes have greater activation**. To investigate whether the MAIT cell TCRβ clonotypes that expanded in diagnosed individuals in response to S. Paratyphi A conveyed a functional advantage in their response to antigen, we generated an in vitro model by transducing JRT3 cells with lentiviral vectors containing MAIT cell TCR sequences of interest. We produced lines expressing the most common MAIT TCRα chain TRAV1-2–TRAJ33 (Supplementary Fig. 9) and the MAIT cell TCRβ clonotypes selected from donor 1 and donor 3. The clonotypes chosen were those that had expanded from pre- to post-infection by a large fold-change and compared this with a clonotype that had contracted by a large fold-change and used the most common TCRβ chain (TRBV20-1). We tested their activation in response to bacterial supernatant and riboflavin metabolite ligands. Details of the TCRβ clonotypes selected are

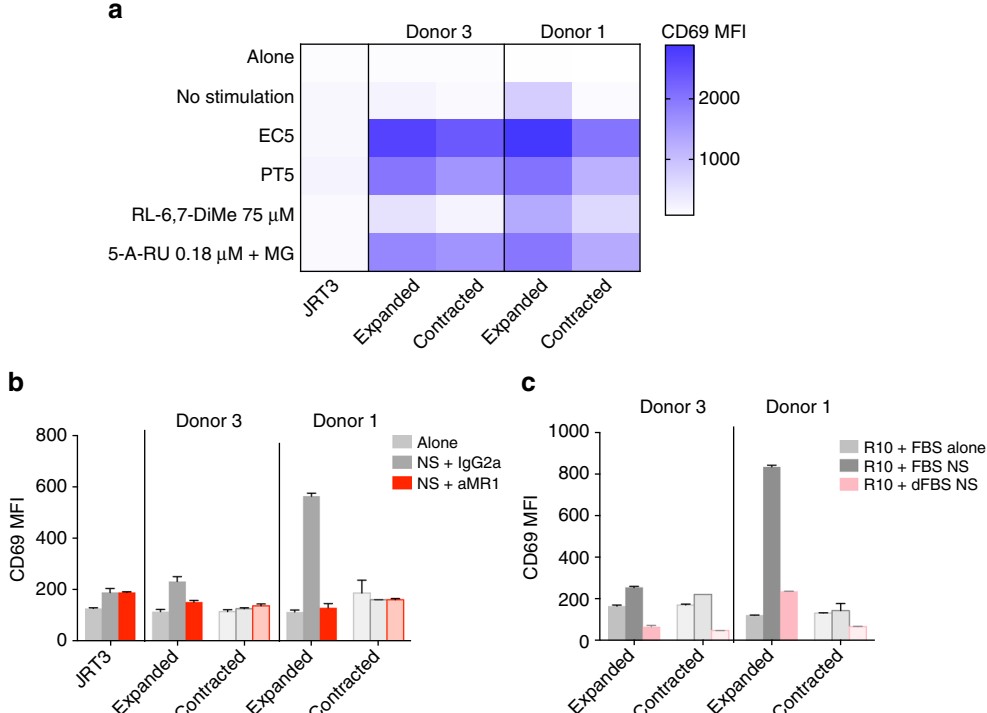

**Fig. 5** Greater activation of Jurkat.MAIT cells expressing TCRβ clonotypes expanded after enteric fever. Jurkat.MAIT cell lines expressing the TCRβ chain of clonotypes that either expanded or contracted in diagnosed donor 1 and donor 3 were produced. Using C1R.MR1 cells as antigen-presenting cells, the activation of Jurkat.MAIT lines was measured by flow cytometry using the MFI of CD69 in response to in vitro bacterial supernatant and ligand stimulation when co-cultured for 18–20 h. **a** Heatmap of CD69 expression on Jurkat.MAIT cell lines when co-cultured with 5 μL *E. coli* (EC) supernatant, 5 μL *S.* Paratyphi A (PT) supernatant, 0.18 μM 5-amino-6-(D-ribitylamino)uracil (5-A-RU) with 10 μM methylglyoxal (MG) and 75 μM 6,7-dimethylribityl lumazine (RL-6,7-DiMe). **b** Baseline activation of Jurkat.MAIT cell lines when co-cultured without stimulus added and **c** with normal vs. dialyzed FBS. For blocking experiments, 10 μg/mL of MR1 blocking antibody (aMR1) or IgG2a isotype control was added. Bars show mean with error bars representing SEM between technical duplicates. Results are representative of at least two independent experiments. MFI, mean fluorescence intensity; NS, no stimulation

presented in Table 1. The Jurkat.MAIT gating strategy, Vα7.2 expression and CD3 mean fluorescence intensity (MFI) overlay (ensuring comparable levels of surface TCR across the Jurkat.MAIT line pairs) are shown in Supplementary Figs. 10 and 11, respectively.

*Escherichia coli* has been previously shown to produce MAIT cell-activating ligands through the riboflavin synthesis pathway[26]. As *S.* Paratyphi A also contains the riboflavin synthesis pathway they would be expected to also produce MAIT cell-activating ligands. In response to either *S.* Paratyphi A or *E. coli* overnight culture supernatant, the expanded TCRβ clonotypes showed greater activation compared to the contracted clonotypes (Fig. 5a). All lines displayed a dose-dependent response and their activation could be completely blocked with the use of an MR1 blocking antibody (Supplementary Fig. 12). This suggests that the expanded TCRβ clonotypes have greater activation, but this is not an exclusive response to *S.* Paratyphi A, as the clonotypes do not convey preferential specificity between the different bacteria.

To investigate whether the extent of Jurkat.MAIT activation to bacterial supernatant by the different TCRβ clonotypes correlated to their activation by stimulatory riboflavin metabolite ligands, we stimulated these cells with the strong precursor MAIT ligand 5-amino-6-(D-ribitylamino)uracil (5-A-RU) plus methylglyoxal and the weaker ligand 6,7-dimethylribityl lumazine (RL-6,7-DiMe) (Fig. 5a). Both lines showed a dose-dependent and MR1-driven response to the ligands (Supplementary Fig. 13). However, the expanded TCRβ clonotypes showed a greater activation overall by both ligands when compared to the contracted clonotype, particularly for donor 1 (which was further confirmed by studying the rate of dissociation of 5-(2-oxopropylideneamino)-

6-D-ribitylaminouracil (5-OP-RU) MR1 tetramers bound to Jurkat.MAIT lines in the presence of excess anti-MR1 blocking antibody, shown in Supplementary Fig. 14). This pattern of activation reflects what we observed in the bacterial supernatant experiments, suggesting that the riboflavin derivatives were the primary antigen that the Jurkat.MAIT lines were responding to in the bacterial supernatant.

When co-culturing the Jurkat.MAIT lines with C1R.MR1 cells without addition of bacteria or ligand, we observed CD69 expression on the expanded Jurkat.MAIT line from donor 1 was fivefold higher compared to when cultured alone (Fig. 5b). This baseline activation, that can be blocked by MR1 blocking antibody, suggests that this clonotype was activated by components in the cell culture media. This was determined to be due to the vitamin metabolites present in the FBS, as the use of dialyzed FBS, which fails to alter activation of HLA-restricted T cells (Supplementary Fig. 15), reduced this baseline response (Fig. 5c).

Together, these results suggest that certain MAIT cell clonotypes may expand in response to infection due to their greater functional avidity of their TCR to MAIT cell ligands.

## Discussion

MAIT cells and their bacterial reactivity are conserved across species, indicating their vital role in the host immune response to bacterial infections[6]. In this study, we had a unique opportunity to further our understanding of MAIT cells and their clonal response in bacterial infections as we utilized samples from a controlled human infection model of the invasive bacterial pathogen *S.* Paratyphi A in healthy volunteers. This has provided

a powerful tool to study how the MAIT cell response progresses during a systemic bacterial infection.

We observed that the relative frequency of circulating MAIT cells significantly decreased early in individuals that developed enteric fever. This could indicate that, as early as day 4, the MAIT cells in these individuals are responding to the presence of S. Paratyphi A in the host. This migration of MAIT cells from circulation has been suggested previously by studies reporting lower frequency of circulating MAIT cells in infected patients compared with healthy individuals[16,28]. In addition to this, a decrease in circulating MAIT cell frequencies was observed in trials that administered live S. Typhi[23] and live-attenuated Shigella dysenteriae 1 vaccine[17]. The findings of these previous studies are in line with our observations, and suggest that circulating MAIT cells are responsive to in vivo bacterial infection and may decrease in frequency in the blood as they move to locally inflamed and infected tissues. Accumulation of MAIT cells in infected tissues has been demonstrated by studies using lung infection mouse models[8,18]. We also observed a recovery in MAIT cell frequency following antibiotic treatment, suggesting that MAIT cell redistribution and reduction in circulating numbers is reversible in bacterial infection. This observed MAIT cell recovery is unlike the non-reversible decrease of MAIT cells that is observed in human immunodeficiency virus (HIV) infection[29,30].

We show that a proportion of circulating MAIT cells in diagnosed individuals were activated and proliferating at the peak of infection, indicating that they were playing a role in the host response to S. Paratyphi A. Similar observations have been made in studies assessing MAIT cell responses in vaccination and live infection trials[17,23,24]. However, the mode of activation of MAIT cells has not been addressed in these human in vivo studies. This is an important point as in vitro studies have shown that MAIT cells can become activated both through TCR stimulation or through TCR-independent cytokine stimulation[19,31], which we also observed after in vitro S. Paratyphi A infection.

To address whether the activation we observed was due to the MR1–TCR interaction, and thus due to the presence of bacteria-derived ligand, we assessed the MAIT cell TCR repertoire and found that, although the TRBV maintained oligoclonality, the TCR repertoire had undergone reshaping and clonal contraction/expansion at the clonotypic level during infection. This suggests that the TCR–MR1 interaction may be influencing the circulating MAIT cell population. There are several studies which have indirectly provided evidence that the TCR–MR1 interaction is important for MAIT cell responses to bacterial infection. In a study of critically ill patients, the circulating MAIT cell proportion in patients with streptococcal infections (bacteria which lack the riboflavin synthesis pathway) was higher than those with infections by bacteria which do produce riboflavin metabolites[28]. Also, a recent study using S. enterica serovar Typhimurium in a mouse lung infection model demonstrated that the MAIT cell response to infection required MR1 and the ability for bacteria to produce the riboflavin-derived ligand[18]. These results support the concept that the riboflavin metabolite ligands are important for the MAIT cell response to bacterial infection and that the changes we observed in TCR repertoire could be driven by the riboflavin metabolite antigen.

We observed an increased functional avidity of expanded MAIT cell clonotypes to the riboflavin ligand compared with a contracted clonotype from the same diagnosed individual. The increased functional avidity could, in part, explain the selective clonal expansion we observed in diagnosed individuals. The concept of MAIT cell selectivity based on TCR sequence has been suggested previously by a study assessing MAIT cell TCR repertoire after in vitro infection[14]. However, this study also suggested differences in TRBV chain usage as well as suggesting ligand/pathogen discrimination, features which we did not observe in our study. Our results showed that the Jurkat.MAIT activation level differed between clonotypes, but that this activation was relatively conserved, regardless of whether they were stimulated with S. Paratyphi A, E. coli, or different riboflavin metabolites. Our observations are supported by the findings from a MAIT TCR–MR1 structural biology study that demonstrated that the CDR3β region directly interacts with the MAIT cell ligand and could influence and "fine-tune" the MAIT cell response[15].

It is likely that the increase in TCR functional avidity for riboflavin metabolites is not exclusively the driving factor for preferential expansion of certain MAIT clonotypes. A driving factor, which we were not able to explore, was the cytokine-dependent activation that we demonstrated in the in vitro S. Paratyphi A challenge assay. Recent advances in methodology that can couple TCR sequence with transcriptional profiling of single cells[32] could help elucidate this, along with other factors that could contribute to MAIT cell clonal expansion in response to infection.

In summary, we have determined the timing and dynamics of the MAIT cell response to S. Paratyphi A infection. We have clearly shown that MAIT cells undergo an adaptive clonal response during infection, yet are still driven by their specificity to a narrow set of vitamin metabolite antigens. These findings demonstrate how MAIT cells sit directly at the interface between innate and adaptive immunity and that lifetime exposure to different microbial infections could impact and shape the host's MAIT cell TCR repertoire. To understand the functional implications of this, future experiments should assess whether select MAIT cell clonotypes that expand in response to S. Paratyphi A infection may play a protective role in subsequent infections with S. enterica or other bacterial species.

## Methods

**Participants and human infection model**. The controlled human S. Paratyphi A infection model utilized has been previously described[25]. Briefly, adult participants (aged 18–60) ingested a single dose of $1–5\times10^3$ colony-forming units (CFU) of S. Paratyphi A strain NVGH308 suspended in $NaHCO_3$ (0.53 g/30 mL) solution. The following 14 days involved daily follow-up and blood monitoring. A diagnosis of enteric fever was defined as S. Paratyphi A bacteremia and/or persistent fever (>38 °C) lasting >12 h. Participants were treated with ciprofloxacin (500 mg, twice daily for 14 days) at the time of diagnosis, or at day 14 if they remained asymptomatic and were not diagnosed. Healthy adult blood donor controls were from laboratory volunteers or leukocyte cones obtained from NHS blood and transplant service. Peripheral blood mononuclear cells (PBMCs) were isolated from all blood samples and leukocyte cones by gradient centrifugation using Lymphoprep™ (AxisShield).

**Cell lines and transfection**. C1R and J.RT3-T3.5 (JRT3) cell lines (ATCC) were cultured in RPMI-1640 supplemented with 10% fetal bovine serum (FBS), 1% penicillin-streptomycin and 2 mM L-glutamine. The 293T cell line (ATCC) was cultured in DMEM supplement with 10% FBS, 1% penicillin-streptomycin and 2 mM L-glutamine. All cell lines were tested negative for mycoplasma contamination.

The C1R.MR1 cell line was produced by cloning the MR1 gene into a pHR-IRES-GFP vector which was then co-transfected into 293T cells with the HIV gag-pol and VSV-G expression plasmids using FuGENE 6 Transfection Reagent (Roche) according to the manufacturer's instructions. The supernatant from this culture containing the lentiviral particles was then used to transduce C1R cells that were then sorted based on high expression of GFP. MR1 expression was then confirmed by MR1 antibody staining and flow cytometry.

The Jurkat.MAIT cell lines were produced by cloning into a pHR-IRES vector the TRAV1-2–TRAJ33 (CDR3α: CAVMDSNYQLIW) MAIT cell TCRα chain and then cloning in the TCRβ chain of interest. The vector was then co-transfected into 293T cells with the HIV gag-pol and VSV-G expression plasmids using X-tremeGENE™ 9 DNA Transfection Reagent (Sigma) according to the manufacturer's instructions. The supernatant from this culture containing the lentiviral particles was then used to transduce JRT3 cells (which lack an endogenous TCRβ chain). TCR expression and pairing with MAIT cell TRAV1-2 (Vα7.2) chain was confirmed by flow cytometry and cells sorted based on CD3 and Vα7.2 expression.

**Table 2 Primers for TCR repertoire amplification**

| PRIMER NAME | PRIMER SEQUENCE (5′-3′) | USE |
|---|---|---|
| AC1R_RV | ACACATCAGAATCCTTACTTTG | TCRα |
| AC2R_RV | TACACGGCAGGGTCAGGGT | TCRα |
| BC1R_RV | CAGTATCTGGAGTCATTGA | TCRβ |
| BC2R_RV | TGCTTCTGATGGCTCAAACAC | TCRβ |
| HUM_ACJ_RV | NNNN(XXXXX)GGGTCAGGGTTCTGGATAT | TCRα |
| HUM_BCJ_RV | NNNN(XXXXX)ACACSTTKTTCAGGTCCTC | TCRβ |
| HUM_STEP1_FW | NNNN(XXXXX)CACTCTATCCGACAAGCAGT | TCRα/β |
| SMART20_FW | CACTCTATCCGACAAGCAGTGGTATCAACGCAG | TCRβ |
| SMARTTRAV1-2_FW | CACTCTATCCGACAAGCAGTCAGCAACATGCTGGCGAAGC | TCRα |
| SWITCH_OLIGO | AAGCAGTGGTATCAACGCAGAGTACTCTT(rG)₃ | TCRβ |
| TRAV1-2_FW | CAGCAACATGCTGGCGAAGC | TCRα |

XXXXX indicates unique barcoded sequence used for identifying individual samples after pooling. Primers sequences based on those previously published[35]

**Preparation of bacterial stocks.** All work involving *S.* Paratyphi A was carried out in a CAT3 facility. *S.* Paratyphi A strain NVGH308 and *E. coli* strain DH5α were plated on Lysogeny broth (LB) agar plates overnight at 37 °C. A single colony was picked, plated, and incubated overnight at 37 °C to ensure single origin colony. A single colony was chosen and grown in LB+10% sucrose overnight. LB+10% sucrose was inoculated with the overnight culture 1:100 and grown for 3 h (to log-phase growth point). Frozen aliquots were stored at −80 °C and thawed just prior to use in in vitro experiments. The CFU for each bacterial stock was determined and used for all multiplicity of infection calculations.

**In vitro infection.** For co-cultures, C1R.MR1 and Jurkat.MAIT cells (at a ratio of 2:1) were incubated with supernatant from an overnight bacterial culture grown in LB. For ligand experiments, RL-6,7-DiMe (Toronto Research Chemicals) or 10 µM methylglyoxal with 5-A-RU[26] was added directly to C1R.MR1 cells. Cells were then incubated for 18–20 h.

For PBMCs, cells were infected with live *S.* Paratyphi A strain NVGH308 or *E. coli* strain DH5α from frozen mid-log phase stocks. After 1 h, 100 µg/mL gentamicin was added and samples left for 16–18 h. For intracellular staining, 5 µg/mL brefeldin A solution was then added and incubated for 6 h. Intracellular Fixation & Permeabilization Buffer (eBioscience) was used for intracellular staining.

For MR1 blocking experiments, 10 µg/mL anti-MR1 antibody (clone 26.5, Biolegend) or mouse IgG2a isotype control (MOPC-173, Biolegend) were used. For IL-12 blocking experiments, LEAF™ purified anti-human IL12/23 p40 antibody (C11.5, Biolegend) and LEAF™ purified mouse IgG1 isotype control (MG1-45, Biolegend) were used. Antibodies were pre-incubated for 1 h with C1R.MR1 cells or PBMCs prior to addition of bacteria, supernatant, or ligand.

**Flow cytometry and cell sorting.** Antibodies for flow cytometry and cell sorting were from Biolegend unless otherwise indicated. Antibodies were: CD161 APC (HP-3G10), CD3 PE/Cy7 (UCHT1), CD3 APC/Cy7 (HIT3a), CD38 CF594 (HIT2, BD Biosciences), CD4 Alexa Fluor® 700 (RPA-T4), CD69 Pacific Blue™ (FN50), TCR γ/δ BV421™ (B1), Granzyme B PE (GB11, BD Biosciences), IFN-γ BV786 (4S.B3, BD Biosciences), TCR Vα7.2 BV605™ or PerCP/Cy5.5 (3C10), TNF BV650 (MAb11, BD Biosciences). Dead cells were excluded using LIVE/DEAD® Fixable Aqua Dead Cell Stain Kit (Life Technologies). Flow cytometry data was collected on BD LSRFortessa™ X20 or Invitrogen Attune NxT. Cell sorting was performed on BD FACSAria™ Fusion Cell Sorter or BD FACSAria™ III. Data was analyzed using FlowJo™ cell analysis software (FlowJo, LLC).

**Tetramer staining and decay assay.** The MR1 5-OP-RU and 6-formylpterin (6-FP) PE-conjugated tetramers were provided by the NIH Tetramer Core Facility. For staining PBMCs, cells were stained with LIVE/DEAD® Fixable Aqua Dead Cell Stain Kit, followed by incubation with the MR1 tetramer (5 µg/mL) for 40 min at room temperature followed by surface antibody staining for 20 min at 4 °C. For the tetramer decay assay, Jurkat.MAIT lines were stained with LIVE/DEAD® Fixable Aqua Dead Cell Stain Kit followed by incubation with 5-OP-RU MR1 tetramer (2.5 µg/mL) for 40 min at room temperature. Excess tetramer was washed off and cells re-suspended in buffer with or without 20 µg/mL anti-MR1 antibody (clone 26.5, Biolegend). Samples were left at 37 °C and periodic samples taken, washed and fixed immediately. Once all samples were collected, CD3 surface staining was performed.

**Mass cytometry sample preparation.** Frozen PBMCs (2–3 million) from challenged volunteers were thawed and washed with RPMI-1640 media supplemented with 1% FBS, 1% penicillin/streptomycin, 2 mM L-glutamine, 1% 1 M HEPES and 50 µM β-mercaptoethanol. Cells were washed twice in cold PBS, and incubated on ice with 200 µM cisplatin for 5 min. Cells were then washed with CyFACS buffer (PBS+4% FBS+0.05% sodium azide) and stained with α-GalCer-Eu153 tetramer (that was produced from α-GalCer monomers that were tetramerized using streptavidin conjugated to 153Eu) for 30 min at room temperature in the presence of 10 µM free biotin. This was followed by a 30-min incubation in primary antibody mix (Supplementary Table 1) on ice. Cells were then washed twice in CyFACS and stained with metal-tagged surface antibodies (Supplementary Table 2). After 30 min, the cells were washed twice with CyFACS and incubated in Foxp3 Fixation/Permeabilization buffer (eBioscience) on ice for 30 min to enable intranuclear staining. Cells were then washed twice in Permeabilization (perm) buffer (Biolegend) (as this is the permeabilization buffer optimized for our CyTOF protocol) and stained with metal-tagged intracellular/intranuclear antibodies (Supplementary Table 3) for 30 min on ice. After washing twice in perm buffer, cells were incubated on ice with metal-tagged streptavidin-155 for 10 min. Cells were then washed twice in perm buffer, once in PBS, and fixed in 2% paraformaldehyde (PFA) at 4 °C. The following day, cells were washed twice with perm buffer and once with PBS.

**Mass cytometry sample barcoding.** We next barcoded stained samples to eliminate doublets and be able to run multiple samples simultaneously. Cells were incubated with cellular barcodes for 30 min as previously described[33]. Pt-102, Rh-103, Pd-104, -105, -106, -108, and -110 were used for this experiment. 113 was not used due to interference with CD57 on mass 115, based on our observations from previous experiments. A small degree of interference was observed between 110 and 112/114 but this did not affect de-barcoding as we could easily gate on 110 positive cells without including 112/114 positive cells. In addition, since CD14-positive cells (112 and 114 positive) were excluded for analysis, interference between 110 and 112/114 was not an issue.

**Mass cytometry sample batching and acquisition.** Cells were then washed once with perm buffer and incubated in CyFACS for 10 min on ice. Cellular DNA was labeled at room temperature with 250 nM iridium interchelator (Fluidigm) diluted in PBS with 2% PFA. After cellular DNA labeling cells were washed twice with CyFACS and kept in a 96-well U-bottom plate.

To accommodate the required numbers of samples, two barcoded batches were prepared, ensuring each batch included all time-points for a given volunteer. A single healthy donor's PBMCs, that was prepared and stained in parallel with volunteer samples, was included in each batch as a control for batch-to-batch variation.

For acquisition, small aliquots of cells from each batch were filtered through a 0.35 µM cell strainer into a 5 mL polystyrene tube. The batch aliquot was then washed twice in water before being resuspended in water at 500,000 cells/mL. EQ Four Element Calibration Beads (Fluidigm) were added at a final concentration of 1% prior to sample acquisition. The same pooling steps were repeated each time the acquisition of an aliquoted batch was completed. This approach was taken to minimize the duration the stained cells were kept in water (1–2 h maximum).

Both batches of samples were acquired on subsequent days to minimize batch effects which can occur if stained cells are stored in CyFACS long term. Batch effect was analyzed by performing a tSNE plot to compare the PBMCs from the healthy donor stained and acquired with the two distinct batches. Cells acquisition was performed on a CyTOF2 (Fluidigm) and the total number of live cells acquired for each sample was between 30,000 and 100,000.

**Mass cytometry data analysis.** After mass cytometry acquisition, the data was exported in flow cytometry file format, normalized[34] and events with parameters having zero values were randomized using a uniform distribution of values between −1 and 0. Each sample containing a unique combination of two metal barcodes was

de-convoluted using manual gating in FlowJo to select cells stained with two (and only two) barcoding channels. MAIT cells were identified as CD14⁻ CD19⁻ CD3⁺ Vα7.2⁺ CD161⁺ and marker expression assessed using Cytobank (MRC) software. The total number of MAIT cells identified in each sample ranged from 83 to 346.

**Flow cytometry for validating mass cytometry experiments**. Samples used for mass cytometry validation were 200 μL of whole blood that were fixed in 1-step Fix/Lyse Solution (eBioscience), and stored at −80 °C. Fixed cells were thawed, permeabilized using Permeabilization Buffer (eBioscience) and then stained with the following antibodies specific for epitopes insensitive to fixation, purchased from Biolegend unless otherwise indicated: CD38 BB515 (HIT2, BD Biosciences), CD4 BV510™ (SK3), CD8 PerCP/Cy5.5 (HIT8a), CD161 APC (HP-3G10, eBioscience), HLA-DR APC/Cy7 (L243), TCR Vα7.2 BV605™ (3C10), TCR γ/δ BV421™ (B1), Ki67 PE (Ki-67), and CD3 BV650™ (UCHT1).

**Activation assay for assessing dialyzed FBS**. 100,000 CD4⁺ T cells specific for the *S. enterica* protein CdtB were stimulated with 3 μg/mL CdtB 105−125 peptide for 18 h in culture media containing either normal or dialyzed FBS (Gibco). Cells were then stained for flow cytometry using anti-CD69 (A488)-conjugated antibody.

**TCRβ repertoire library preparation**. Cells for TCR repertoire analysis were sorted directly into lysis buffer from RNAqueous®-Micro Total RNA Isolation Kit (Ambion). Sorted cell numbers ranged from 5000 to 10,000 cells per sample. RNA was then extracted using the RNAqueous®-Micro Total RNA Isolation Kit following the manufacturer's instructions. The unbiased TCR amplification method was used as previously described[35]. Briefly, template-switch cDNA was produced using SMARTScribe™ Reverse Transcriptase (Clontech) with switch_oligo and BC1R primers described in Table 2. First and second PCR steps were conducted using Phusion® High-Fidelity DNA Polymerase (New England Biolabs) and primers described in Table 2. Samples were pooled and NEBNext® Ultra™ DNA Library Prep Kit for Illumina® (New England Biolabs) was used following the manufacturer's instructions. Samples were sequenced on Illumina MiSeq platform using MiSeq reagent kit V2 300 cycle (Illumina).

**TCRα repertoire library preparation**. cDNA was synthesized using the oligo-dT primer method of the RETROscript® Kit (Ambion) following the manufacturer's instructions. The TCRα chain was amplified by PCR using Phusion® master mix and primers specific for TRAV1-2 (TRAV1-2_FW) and the TRAC region (AC1R_RV) primer details outlined in Table 2. The PCR products were used for a second nested PCR using primers Smart_TRAV1-2_FW and AC2R_RV. PCR product was used as a template for a third nested PCR using primers Step_1_FW and Hum_bcj_RV containing a barcoded sequence. Samples were then pooled for MiSeq analysis as described in TCRβ library preparation.

**TCR repertoire analysis**. The sequencing files generated by the MiSeq were split based on their barcoded sequence using Jemultiplexer (GBCS) and then run through the MiTCR software[36], aligning TCRα/β sequences using default parameters. VDJ Tools[37] was then used to filter out the non-functional sequences prior to post-analysis visualization. Clonotypes were defined based on their amino acid sequence. Frequencies were determined by number of assigned sequence reads. Default strict sample intersection rules were applied to VDJ tools analysis.

**Statistical analysis**. Statistical analysis was performed using Prism software (Graphpad). For multiple comparisons, one-way ANOVA was used with Dunnett's test to correct for multiple comparisons. For comparison between two groups, either Student's paired two-tailed *t*-test or unpaired two-tailed *t*-test was used. Significant result is indicated where $*P < 0.05$, $**P < 0.01$, $***P < 0.001$, $****P < 0.0001$. Sample sizes were limited by the availability of volunteer samples but adequate to detect differences between sample groups, as indicated by the reproducibility and variability of each experiment.

**Ethics**. Written informed consent was obtained for all study volunteers in accordance with the Declaration of Helsinki and ethical approval was received from Oxford Research Ethics Committee (14/SC/0004 and 14/SC/1204). Samples obtained from the controlled human *S. Paratyphi A* infection model were part of registered clinical trials (NCT02100397 and NCT02192008) and conformed to the Good Clinical Practice (GCP) guidelines.

**Data availability**. Sequence data that support the findings of this study have been deposited in NCBI Sequence Read Archive with the primary accession code PRJNA412739.

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

## Acknowledgements

We would like to thank the study volunteers. Thank you to Rosanna McEwen-Smith, Victoria Woodcock, and Andreas V. Hadjinicolaou for assistance with blood collection, Craig Waugh for assistance with cell sorting, Tim Rostron for assistance with next-generation sequencing, Mike Stubbington for reading and editing the manuscript and Hashem Koohy for assistance with statistical analysis. The MR1 tetramer technology was developed jointly by James McCluskey, Jamie Rossjohn, and David Fairlie, and the material was produced by the NIH Tetramer Core Facility as permitted to be distributed by the University of Melbourne. This work was supported by the UK Medical Research Council (MRC) (MR/K021222/1), Cancer Research UK (CRUK) (C399/A2291), the Oxford National Institute for Health Research (NIHR) Biomedical Research Centre, the Bill & Melinda Gates Foundation (OPP1084259), the European Vaccine Initiative (PIM) and core funding from the Singapore Immunology Network (SIgN) and the SIgN immunomonitoring platform. G.S.B. acknowledges support in the form of a Personal Research Chair from Mr. James Bardrick and the UK MRC (MR/K012118/1).

## Author contributions

L.J.H. conceived and designed the study, processed the samples, performed the experiments, analyzed the data, and wrote the manuscript. G.N. conceived and designed the study, processed the samples, performed the flow cytometry experiments, analyzed the data, and edited the manuscript. D.S. designed and performed cellular assay experiments. H.G. conceived and designed TCR repertoire experiments. P.K. processed the samples and performed the flow cytometry experiments. M.R. designed and conducted cell sorting experiments. L.P.-L. conceived the in vitro infection model and provided bacteria handling training. K.W.W.T. and E.W.N. performed the mass cytometry experiments. H.C.D., M.M.G. and A.J.P. conceived the human challenge model, obtained funding, and directed the clinical trial and sampling. N.V. and G.S.B. synthesized the 5-A-RU compound. V.C. conceived and designed the study, edited the manuscript and supervised the experiments. All authors discussed the results and commented on the manuscript.

## Additional information

**Competing interests:** The authors declare no competing financial interests.

