## [Peer Review File · Nature Communications]

Reviewers' comments:

Reviewer #1 (MAIT)(Remarks to the Author):

Summary

The study characterises the MAIT cell response and TCR repertoire following *S. Paratyphi* infection in human volunteers. This is the first study of this type with matched samples of pre and post infection in humans, and is particularly valuable given the variation in MAIT cell numbers between healthy humans.

The authors show that MAIT cells are decreased in the blood early after infection, are activated, and then recover to higher numbers than pre-infection. There is clonal expansion and a change in repertoire following infection, and expanded clones are not pathogen-specific, but reactive to metabolites and other bacteria.

MAIT cells recognise and respond to riboflavin metabolites produced by a wide range of organisms. The protective role of MAIT cells infection has been demonstrated in mouse models for a limited number of pathogens, and most human data are almost necessarily correlative. To some degree this means that "vaccination" only makes sense if MAIT cells are protective and yet this is challenging to demonstrate in humans. Nonetheless, there are remaining questions in the field surrounding the full range of antigens recognised by MAIT cells and whether pathogen-specific responses occur to unique or particular antigens. Hence, this study reveals important information, potentially relevant to many bacterial infections.

Main concerns

The conclusion that MAIT cells were decreased in the blood of infected patients is shown Figure 1. Here, there is a significant difference between blood MAIT cell numbers in naïve versus day 7-8 donors diagnosed with enteric fever. This difference was non-significant in non-diagnosed controls. However, the magnitude of the decreased cell numbers appears comparable in non-diagnosed despite not being statistically significant. Differences are more compelling when activation markers are studied between the two groups, but I wonder if the authors confident that there are real differences in MAIT cell depletion in the blood in those diagnosed or not with enteric fever?

Since it is the main point of the paper, the interesting and provocative observation that the TCR repertoire is altered by infection needs sharper justification, particularly given the low sample numbers. What would be the spread of data with technical replicates on the same sample with this method? Is there a statistical test to show that the diagnosed infected individual's repertoires are different to those observed pre-infection. Is there a comparable change in the controls? Looking at Figure 2B and S5, at least 3 of the donors look similar to the controls.

The comparison of "expanded" and "contracted" TCR sequences is examined in the in vitro Jurkat system. A functional comparison of "expanded" and "contracted" TCR sequences is examined in the Jurkat system. The authors are diligent in controlling for TcR expression levels, but the analysis really needs larger sample numbers to be statistically compelling. The conclusions are plausible within the parameters of the limited analysis, but currently only one clone for each is compared. However, this is insufficient to generalise the conclusion. The assertion that some emergent clones have high "functional avidity" would be strengthened with some additional experiments, such as tetramer "wash-off" to give an indication of relative binding strength to MR1-Ag. Do the MAIT cells stain more brightly as a population after infection than in pre-infection samples? Is it clear that co-receptor expression is properly controlled (e.g. CD8) in the comparisons between "expanded" and "contracted" clones, when measuring "functional avidity"?

Referencing is insufficient in a few cases:

Page 3 - In addition to *F. tularensis*, mouse models of other infection including *Klebsiella pneumoniae* and *Mycobacterium tuberculosis* also show a role for MAIT cells and these studies should also be referenced.

Page 3 - MR1 independent activation of MAIT cells – could include Kurioka et al. reference in addition to Ussher et al.

Page 6 – activation by 5-A-RU + methylglyoxal was first described in a paper by Corbett et al. Nature 2012. In addition, the concentration of methylglyoxal is lacking from the methods.

Page 10 – Reantragoon 2013 J. Exp Med could also be included with the Lepore reference.

Page 10 – The claims on MR1 partially blocking the response are valid, but seem to be forgotten in later sections where only the TCR-dependent stimulation is considered. Additionally, it would be valuable to understand whether blocking with a higher dose of MR1 considered (10 ug/ml was used). Anti-IL-12 also blocked only partially. Would a combination of the two give full blockade of signal? Given the authors claim both antigen-driven and cytokine-dependent activation, how does this relate to the TCR clonal expansion seen?

Page 13 – The authors suggest the baseline activation could possibly be due to their clone being reactive to folate metabolites. This could be tested in the system used, though I would not consider this the most likely explanation. Are the authors suggesting (as claimed on page 17) that the Jurkat lines are all reactive to folate metabolites as well as the riboflavin metabolites that expanded these clones during infection? These Folate-reactive cells would be expected to be autoreactive based on the known structural data. Are they folate reactive and expanded in a cytokine-driven response? Do the CDR3beta sequences match those in the referenced paper?

Minor comments and typographical errors:

Page 2 – “infection led to a transient contraction of over-represented clones” is confusing. Perhaps better stated as “at the peak of infection there was transient contraction of clones that were over-represented prior to infection”.

Figure 1 – the time points are not clear? At what point was diagnosis made after challenge? Was this the same time point in all cases? This should be indicated in the figure.

Figure 1 (and other figures). The use of D for “donor” could be confused with “day” after infection. Perhaps better to label the figure as “Donor 1” etc.

If Supplementary Figure S4 could be incorporated into Figure 2, this would better represent the whole data set than showing one donor only.

Page 13 – “*S. Paratyphi* and *E.coli* are both known to have the riboflavin pathway and therefore can produce riboflavin metabolite ligands.” Should be more accurately worded. In fact *E.coli* ligands have been demonstrated to stimulate MAIT cells and to produce MAIT-activating ligand (Corbett et al. Nature 2014). *S. Paratyphi* “would be expected to produce....”

Page 15 - Was the increase in MAIT cells with antibiotic treatment due to the treatment, or just due to recovery over time? Can these possibilities be separated?

Page 16 – “*S. typhimurium*” should be “*S. Typhimurium*.”

Do the authors see downregulation of CD3 on active clones (and thus their loss from analysis?)

Were other markers of activation considered in addition to CD38?

Is there a reason why CD4- MAIT cells are excluded? Although they represent only a small proportion of MAIT cells, they could also be included here.

Reviewer #2 (CyToF)(Remarks to the Author):

The below is a technical review concerning mass cytometry experiments only as requested by the Editor for the manuscript entitled "A human infection model of live Salmonella Paratyphi A demonstrates clonal expansion of MAIT cells and shaping of their TCR repertoire", authors: Howson L.J. et al.

Mass cytometry materials and methods are summarized in supplementary methods, data are shown in figure S2 and stated in lines 175-182 of the results section. A 42 marker panel plus barcodes, Iridium stain and cis-platin was used to analyze 4 patient samples plus one healthy donor control.

General comment: The supplementary methods section describing mass cytometry experiments is written to some extent in a confusing and incomplete way and requires reworking.

Points in detail on supplementary methods, headline Mass Cytometry

- (1) "Frozen cells were thawed": please specify which cells and the starting cell number for mass cytometry staining.
- (2) "RPMI supplemented with... 1x beta-mercaptoethanol": 1x is not a concentration, please specify. Also, please specify why this supplement is necessary as it is not a standard component in thawing medium for what seems to be PBMCs.
- (3) "Cells were pre-stained... and left untreated at 37 C for 4 hrs in the presence of monensin and brefeldin": Please explain why the cells are left in the presence of a protein transport inhibitor cocktail which is usually added in case of stimulation protocols but not for straight stains.
- (4) "... stained with streptavidin α -GalCer". Table S1 specifies that this reagent is on metal mass 153: specify if this is a biotin-streptavidin dual-step reagent or how it is structured and stained for. As the panel contains a second Streptavidin reagent in FoxP3 on metal 155 it would be important to understand how this stain is done exactly and to show background controls for GalCer and FoxP3 to exclude there has been no double staining of the GalCer.
- (5) "This was followed by a 30 minute incubation in primary antibody mix on ice. Cells were then washed twice in CyFACS ad stained with metal-tagged surface antibodies. After 30 minutes...": I assume one of these stains has not happened as the surface cocktail is normally not added twice, please correct/elaborate.
- (6) "... incubated in FoxP3 Fixation /Permeabilization (eBiosciences) buffer on ice for 30 minutes. Cells were then washed twice in 1x Permeabilization (perm) buffer (Biolegend) and stained with biotin-anti-human FoxP3..." Clarify why there is a swap from the eBioSciences FoxP3 Fixation/Permeabilization buffer to Biolegend Permeabilization buffer without any antibody stain being done in-between, at least this does not become apparent. The eBiosciences FoxP3 kit has always come complete with Fox-Perm and Perm solution, so a brand swap requires an explanation. Please also specify which intracellular antigens except for FoxP3 and Ki67 were analyzed in this case.
- (7) The authors do not explain when and how they performed the two step staining for the anti-FITC and anti-PE antigens.
- (8) Citation 1 in supplementary methods refers to a paper by Wong et al (Cell reports (2015), 11: 1822) describing to a home-made bar-coding system of RH103, PD 104, 106, 108, 110, 113. Please can the authors specify which of these barcodes were used and how samples were pooled. Please demonstrate that there has been no interference of these barcodes (especially 110/113) with the Q-Dot marker on metals 112/114 and CD57 on mass 115.

(9) Given that barcoding was done at the end of the staining procedures its purpose is to help eliminate doublets (this is not mentioned but should be added) and to run several samples simultaneously, with seemingly 17 total samples run not an absolute necessity.

" To accommodate the required number of samples....Healthy donor was also included in batch as an internal control."

Please can the authors explain how the samples were run/batched, i.e. all samples from one individual plus HD in each case, or HD only once, or ?? How many cells per sample were added into each batch? How did the authors handle cell fragility observed with longer exposure of cells suspended in water for injection? How did the authors control for potential batch effects (if applicable, but unclear if this is a possibility the way this is written).

(10) Please clarify if the healthy donor control was stained in the same way as the patient samples.

(11) Please specify for the mass cytometry runs what number of total cells were acquired and what the total number of live MAIT cells in each measured sample has been given that their frequency is rather low. This will help understand the statistics on page 9 of the results section.

Points in detail on supplementary methods, headline Mass Cytometry data analysis

(12) Mass cytometry data analysis: This section only roughly describes how normalization and debarcoding was done but not how the actual mass cytometry data were analyzed. Please can the authors add this to the methods section.

Points in detail , headlines Results and Supplementary Figures and Tables

(13) Results from line 175 on page 9: The mass cytometry data confirm the observation already made by flow cytometry that CD38 is upregulated on MAIT cells in patients with enteric fever. The data further show an upregulation of Ki67 while none of the remaining markers from the 42 antigen panel showed any significant differences. As this is the only significant mass cytometry result the authors might choose to show the raw data plot of Ki67 vs CD38 at least exemplary for one patient in addition to the statistical workup. As n=4 is a rather low number a follow up on more individuals by regular flow cytometry focusing on CD38/Ki67 would be desirable. Previous publications [e.g Blood (2013), 121(7):1124 or Front Immunol (2017), 8:398] show a correlation of CD38^{hi}/Ki67 positive MAIT cells in HIV as well as after bacterial infection indicating a common trend in marker expression after immune challenge. These papers should be referenced. If the authors think the upregulation of Ki67 as not significantly relevant to the overall findings of the paper they may consider omitting the mass cytometry data altogether.

(14) The authors keep changing the day numbering throughout the paper (e.g in Fig S2a it is 0/4/PD+96/28, in Fig 1 it is 0, 4, 7-8, 9-11, 28...) which is rather confusing and should be standardized for easier reading and comparison of the same type of cell samples.

(15) Figure S2 a and b: Please clarify: in the results section the authors state that cells from 4 infected individuals were analyzed by mass cytometry. Can the authors please clarify what D7/18/19/20 stand for and if all samples were the PBMCs harvested at day 0/4/PD+96 and 28?

(16) The data from the healthy donor control are not shown for comparison in figure S2b, can the authors please include this as a baseline sample or state how these results were used for this manuscript.

Reviewer #3 (unconventional T cells)(Remarks to the Author):

This is a remarkable study where a panel of 25 volunteers were administered live Salmonella Paratyphi and the MAIT cell response tracked through infection and antibiotic treatment. This offers a level of stringency normally only possible with mouse based experiments. This study has allowed the monitoring of MAIT cells throughout the course of infection and resolution and has provided valuable insight into the influence of their TCRb chain on their expansion/contraction

during the course of disease and its impact on their reactivity to antigen in general. I have some comments below:

1. My main concern with the study is that it relies on a surrogate phenotype for MAIT cells (Va7.2 vs CD161 on CD4⁺ T cells). While it is generally accepted that this is a fairly reliable way to identify MAIT cells in most healthy individuals, it is certainly possible that some MAIT cells might downregulate CD161 (as reported during HIV infection) and other non-MAIT cells that are Va7.2⁺ might upregulate CD161. This could easily confound the interpretation of the data in this paper because some MAIT cells might be escaping from the analysis gate while other non-MAIT cells might be moving in to that gate. This would explain why the typical MAIT TCR TRBV genes (6 and 20) are being replaced by other TRBV genes. It was therefore very reassuring at the end of the paper to see the data with Jurkat cells transduced with one representative TCR from the expanded and contracted populations, where they were clearly MR1 dependent. However, only one TCR being tested does not really negate the concern that many or most of these cells that become increased after infection might not be MAIT cells. The best way to validate the findings here would be to use MR1-5ARU/MG tetramers to investigate at least some of the key samples, to ensure that the MAIT cells are all still within the gates used, and that there is not an influx of non-MAIT cells in the same gates. The tetramers are available from NIH tetramer facility. If that is not possible, then at least analysis of a larger sample of the cells after infection for MR1 reactivity would be reassuring. For example, the cells could be sorted as a population and cocultured with the CD1-MR1 cells plus antigen and reactivity checked by ICS at the level of individual cells, and blocked with anti-MR1.
2. It is clear that this group has the ability to generate TCR transduced cell lines and this section (Figure 5) provided compelling data on the impact of two different TCRb chains, but unfortunately only one expanded and one contracted TCRb chain was compared. It isn't really possible to derive general conclusions from n=1 sample and this would be much better if 2-3 or more of each TCRb type (expanded vs contracted) was compared.
3. The focus on the TCR changes is on TRBV genes, but human MAIT cells can also vary for TRAJ genes (and possibly also TRAV genes although they would be excluded using the Va7.2 gating approach). It would be useful to determine if the TCR a chain is also modulated during the course of infection.
4. It is not appropriate to say there was a difference that was not statistically significant (p9). The null hypothesis that the groups are the same can't be rejected. However, it looks like the wrong type of ANOVA was used to determine this value. It is appropriate to use a 'repeated measures ANOVA' which would link samples from the same individual. If this was not used it should be because it will give more power to resolve differences within individuals.
5. The IL12 blocking experiments (p10) seem to be overstated. Only 2 donors were tested and only for Salmonella was there an apparent decrease – not for E.coli, and no significance was established in either case.

Point by Point Response to Reviewers' Comments

Reviewer 1

We are very pleased that the Reviewer acknowledges that *“this is the first study of this type with matched samples of pre and post infection in humans, and is particularly valuable given the variation in MAIT cell numbers between healthy humans”* and that *“this study reveals important information, potentially relevant to many bacterial infections”*.

Comment 1:

The conclusion that MAIT cells were decreased in the blood of infected patients is shown Figure 1. Here, there is a significant difference between blood MAIT cell numbers in naïve versus day 7-8 donors diagnosed with enteric fever. This difference was non-significant in non-diagnosed controls. However, the magnitude of the decreased cell numbers appears comparable in non-diagnosed despite not being statistically significant. Differences are more compelling when activation markers are studied between the two groups, but I wonder if the authors confident that there are real differences in MAIT cell depletion in the blood in those diagnosed or not with enteric fever?

Response:

We agree with Reviewer 1 that the decrease that we observe in undiagnosed individuals does appear to be a real difference, and have now added that $P = 0.06$ to **Figure 1a** showing that it was close to significance. We have also re-worded the sentences in this results section referring to the MAIT frequency figures to better convey that the main difference observed between diagnosed and undiagnosed individuals is the timing in which the decrease occurs (day 4–7 in diagnosed versus day 10 in undiagnosed) which should better clarify the conclusions drawn from these figures (**lines 160–4**).

Comment 2:

Since it is the main point of the paper, the interesting and provocative observation that the TCR repertoire is altered by infection needs sharper justification, particularly given the low sample numbers. What would be the spread of data with technical replicates on the same sample with this method? Is there a statistical test to show that the diagnosed infected individual's repertoires are different to those observed pre-infection. Is there a comparable

change in the controls? Looking at Figure 2B and S5, at least 3 of the donors look similar to the controls.

Response:

This is an important point Reviewer 1 raises regarding **Figure 2** and we agree that there needs to be better clarity in the manuscript as to which figures support the observation that TCR repertoire is altered. We agree that the chain usage alone is not sufficient to support this finding, as the main observation we made was that MAIT cells maintain the oligoclonality of TRBV usage that is observed in the pre-infection samples. As there is no straightforward statistical test to be able to compare the two different clusters of results (diagnosed versus healthy TRBV/TRBJ usage over time) we have reworded this section of the results to be more conservative and use this figure to demonstrate the overall stability of TRBV usage, rather than discussing small changes that cannot be statistically measured (**lines 202–7**).

The repertoire differences in diagnosed individuals are better demonstrated in **Figure 3** and **4**, which show the changes to the repertoire clonotypes, and that this observed difference is able to be statistically measured and is significantly different to what is observed in healthy controls. With regards to the spread of data for technical replicates, we have not included technical replicates in our analysis, but used the healthy controls over time as both biological replicates as well as controlling for changes in repertoire over time, and thus any changes observed in diagnosed individuals that was greater than that in healthy individuals would be attributable to the infection.

Comment 3:

The comparison of “expanded” and “contracted” TCR sequences is examined in the in vitro Jurkat system. A functional comparison of “expanded” and “contracted” TCR sequences is examined in the Jurkat system. The authors are diligent in controlling for TCR expression levels, but the analysis really needs larger sample numbers to be statistically compelling. The conclusions are plausible within the parameters of the limited analysis, but currently only one clone for each is compared. However, this is insufficient to generalise the conclusion.

Response:

We agree with Reviewer 1 and have now increased the sample size to include another expanded/contracted clonotype pair from Donor 3 in **Figure 5** and we have also reworded the conclusions, so they are not over-generalising these results (**lines 280–1**).

Comment 4:

The assertion that some emergent clones have high “functional avidity” would be strengthened with some additional experiments, such as tetramer “wash-off” to give an indication of relative binding strength to MR1-Ag. Do the MAIT cells stain more brightly as a population after infection than in pre-infection samples?

Response:

We agree with Reviewer 1 that the “functional avidity” would be strengthened with a direct TCR engagement experiment. We have included a 5-OP-RU tetramer decay assay for the expanded/contracted donor 1 pair that showed a large difference in activation towards the 5-OP-RU ligand in the cellular assay (**Supplementary Fig. 14**). We also have stained MAIT cells with the tetramer in donor 4 and donor 6 pre-, mid- and post-infection (**Supplementary Fig. 2**) and although they appear to stain more brightly in donor 4, donor 6 (as well as donor 3 and 5 - data not shown) did not show MAIT cells staining more brightly with the tetramer following infection.

Comment 5:

Is it clear that co-receptor expression is properly controlled (e.g. CD8) in the comparisons between “expanded” and “contracted” clones, when measuring “functional avidity”?

Response:

Our J.RT3-T3.5 cell line does not have detectable surface co-receptor expression (CD4/CD8) and even following transduction with functional TCR there is no detectable CD4/CD8 at the surface. Thus, we do not believe co-receptor expression would be contributing to the functional avidity in the Jurkat APC co-culture experiments.

Comment 6:

Referencing is insufficient in a few cases:

*Page 3 - In addition to *F. tularensis*, mouse models of other infection including *Klebsiella pneumoniae* and *Mycobacterium tuberculosis* also show a role for MAIT cells and these studies should also be referenced.*

Page 3 - MR1 independent activation of MAIT cells – could include Kurioka et al. reference in addition to Ussher et al.

Page 6 – activation by 5-A-RU + methylglyoxal was first described in a paper by Corbett et al. Nature 2012. In addition, the concentration of methylglyoxal is lacking from the methods.

Page 10 – Reantragoon 2013 J. Exp Med could also be included with the Lepore reference.

Response:

We are very appreciative for the detailed and critical reading of our manuscript by Reviewer 1 and have incorporated all the suggested additional references into the manuscript.

Comment 7

Page 2 – “infection led to a transient contraction of over-represented clones” is confusing. Perhaps better stated as “at the peak of infection there was transient contraction of clones that were over-represented prior to infection”.

Response:

We agree with this and have changed the wording of this sentence as suggested (**lines 25–7**).

Comment 8:

Figure 1 – the time points are not clear? At what point was diagnosis made after challenge? Was this the same time point in all cases? This should be indicated in the figure.

Response:

We used the color of the symbols (black =before, red = on/after diagnosis) but agree that the exact time of diagnosis is not made clear. We have now added a sentence in the Figure 1 legend outlining that that time of diagnosis was between day 4 and 7 (**lines 489–90**).

Comment 9:

Figure 1 (and other figures). The use of D for “donor” could be confused with “day” after infection. Perhaps better to label the figure as “Donor 1” etc.

Response:

We agree that this could be made clearer, and have now changed all figure keys to label “Donor” rather than “D”.

Comment 10:

If Supplementary Figure S4 could be incorporated into Figure 2, this would better represent the whole data set than showing one donor only.

Response:

We agree and have now incorporated all diagnosed individual’s pie charts into **Figure 2**.

Comment 11:

Page 13 – “S. Paratyphi and E. coli are both known to have the riboflavin pathway and therefore can produce riboflavin metabolite ligands.” Should be more accurately worded. In fact E. coli ligands have been demonstrated to stimulate MAIT cells and to produce MAIT-activating ligand (Corbett et al. Nature 2014). S. Paratyphi “would be expected to produce....”

Response:

We agree and have reworded this sentence to be more clear and accurate (**lines 254–6**).

Comment 12:

Page 15 - Was the increase in MAIT cells with antibiotic treatment due to the treatment, or just due to recovery over time? Can these possibilities be separated?

Response:

This is an important point, however, as both diagnosed and undiagnosed groups are treated with antibiotics, and there is no increase in MAIT cells in the undiagnosed group following treatment, this suggests that antibiotic treatment alone does not influence MAIT cell frequency.

Comment 13:

Page 16 – “S. typhimurium” should be “S. Typhimurium.”

Response:

This has now been changed to the correct nomenclature and capitalisation (**line 321**).

Comment 14:

Page 10 – The claims on MRI partially blocking the response are valid, but seem to be forgotten in later sections where only the TCR-dependent stimulation is considered. Additionally, it would be valuable to understand whether blocking with a higher dose of MRI considered (10 ug/ml was used). Anti-IL-12 also blocked only partially. Would a combination of the two give full blockade of signal? Given the authors claim both antigen-driven and cytokine-dependent activation, how does this relate to the TCR clonal expansion seen?

Response:

We agree that cytokine-driven responses could also influence the MAIT cell response and clonal expansion to *S. Paratyphi A* challenge, and have now added this as a discussion point

(lines 340–2). We have now performed additional blocking experiments to include higher concentrations of MR1 blocking antibody, and combination blocking of both MR1 and IL-12 which resulted in a complete block of activation, see **Supplementary Fig. 5**.

Comment 15:

Page 13 – The authors suggest the baseline activation could possibly be due to their clone being reactive to folate metabolites. This could be tested in the system used, though I would not consider this the most likely explanation. Are the authors suggesting (as claimed on page 17) that the Jurkat lines are all reactive to folate metabolites as well as the riboflavin metabolites that expanded these clones during infection? These Folate-reactive cells would be expected to be autoreactive based on the known structural data. Are they folate reactive and expanded in a cytokine-driven response? Do the CDR3beta sequences match those in the referenced paper?

Response:

We agree that this should be explored further, and we used our experimental system to determine that the source of baseline reactive ligands was the FBS (**Figure 5c**), and that media without folate or riboflavin had very little effect on baseline activation (data not shown). The results and discussion sections have now been amended to reflect this result.

Comment 16:

Do the authors see downregulation of CD3 on active clones (and thus their loss from analysis?)

Were other markers of activation considered in addition to CD38?

Response:

We do not see correlation between expression of CD3 and CD38 on MAIT cells in challenged individuals (as shown in the flow cytometry plots below) and therefore do not believe we are losing these cells from analysis with our gating strategy.

The other activation marker we assessed was HLA-DR, which we did see increase, but only on a small proportion of CD38⁺ MAIT cells.

Comment 17:

Is there a reason why CD4⁻ MAIT cells are excluded? Although they represent only a small proportion of MAIT cells, they could also be included here.

Response:

We agree that a small proportion of MAIT cells can also be CD4⁺ and were omitted by our gating strategy. However, in the interest of having a very strict gating strategy (as at the time of analysis, the MR1 tetramer was not readily available) we wanted to ensure we excluded any non-MAIT cells from analysis. As germline-encoded, mycolyl lipid-reactive (GEM) cells are known to use the TRAV1-2 chain and can express CD161, but are CD4⁺, we chose to exclude CD4⁺ cells from the analysis to reduce possible inclusion of GEM cells in the MAIT cell gate.

Reviewer 2

Comment 1:

"Frozen cells were thawed": please specify which cells and the starting cell number for mass cytometry staining.

Response:

We have now added the information regarding cell type and numbers used for experiments in the **Mass cytometry sample preparation** section of the Supplementary Methods.

Comment 2:

“RPMI supplemented with... 1x beta-mercaptoethanol”: 1x is not a concentration, please specify. Also, please specify why this supplement is necessary as it is not a standard component in thawing medium for what seems to be PBMCs.

Response:

The concentration for beta-mercaptoethanol has now been specified. The addition of beta-mercaptoethanol was done to prevent accumulation of toxic oxygen free-radicals. Since it doesn't cause any negative effects on our PBMCs we use this supplement routinely in all our media (although it is not a standard component in thawing medium).

Comment 3:

“Cells were pre-stained... and left untreated at 37 C for 4 hrs in the presence of monensin and brefeldin”: Please explain why the cells are left in the presence of a protein transport inhibitor cocktail which is usually added in case of stimulation protocols but not for straight stains.

Response:

We apologize for the inclusion of this erroneous statement in the methods as it was not relevant for the staining of this particular panel, as reviewer 2 has correctly pointed out - there are no cytokines in our antibody panel (this was only relevant to the intracellular cytokine staining panels that were run in parallel with these samples but not included in our manuscript).

Comment 4:

“... stained with streptavidin alpha-GalCer”. Table S1 specifies that this reagent is on metal mass 153: specify if this is a biotin-streptavidin dual-step reagent or how it is structured and stained for. As the panel contains a second Streptavidin reagent in FoxP3 on metal 155 it would be important to understand how this stain is done exactly and to show background controls for GalCer and FoxP3 to exclude there has been no double staining of the GalCer.

Response:

We have now added the additional information regarding how the α GalCer was made from monomers were tetramerised using streptavidin conjugated with 153Eu.

Comment 5:

“This was followed by a 30 minute incubation in primary antibody mix on ice. Cells were then washed twice in CyFACS ad stained with metal-tagged surface antibodies. After 30 minutes...”: I assume one of these stains has not happened as the surface cocktail is normally not added twice, please correct/elaborate.

Response:

We agree this was not explained in enough detail and have now added an explanation of the staining protocol which involved first stained the cells with fluorochrome-conjugated antibodies (outlined in **Supplementary Table 1**) followed by incubation with anti-PE and anti-FITC conjugated to metal isotopes together with all the other metal-tagged surface antibodies (outlined in **Supplementary Table 2**).

Comment 6:

“... incubated in FoxP3 Fixation /Permeabilization (eBiosciences) buffer on ice for 30 minutes. Cells were then washed twice in 1x Permeabilization (perm) buffer (Biolegend) and stained with biotin-anti-human FoxP3....” Clarify why there is a swap from the eBioSciences FoxP3 Fixation/Permeabilization buffer to Biolegend Permeabilization buffer without any antibody stain being done in-between, at least this does not become apparent. The eBiosciences FoxP3 kit has always come complete with Fox-Perm and Perm solution, so a brand swap requires an explanation. Please also specify which intracellular antigens except for FoxP3 and Ki67 were analyzed in this case.

Response:

We used the eBioscience fixation/permeabilization buffers to fix and permeabilise the cells to enable identification of intranuclear proteins. The permeabilization buffer from Biolegend was used to wash off the eBioscience fix/perm solution (and used in subsequent staining and washing steps) as it is the buffer, which is used routinely (and has been optimized) for our CyTOF experiments. Hence, we wanted to keep the staining protocol consistent with previous experiments. We have added a clarification of this reasoning in the **Mass Cytometry sample preparation** section of the Supplementary Methods.

Comment 7:

The authors do not explain when and how they performed the two-step staining for the anti-FITC and anti-PE antigens.

Response:

Please see our response to Comment 5 for Reviewer 2.

Comment 8:

Citation 1 in supplementary methods refers to a paper by Wong et al (Cell reports (2015), 11: 1822) describing to a home-made bar-coding system of RH103, PD 104, 106, 108, 110, 113. Please can the authors specify which of these barcodes were used and how samples were pooled. Please demonstrate that there has been no interference of these barcodes (especially 110/113) with the Q-Dot marker on metals 112/114 and CD57 on mass 115.

Response:

Pt-102, Rh-103, Pd-104, -105, -106, -108 and -110 were used for this experiment. 113 was not used due to interference with CD57 on mass 115, based on observations we made from previous experiments. A small degree of interference was observed between 110 and 112/114, but this did not affect de-barcoding as we could easily gate on 110 positive cells without including 112/114 positive cells. In addition, since CD14 positive cells (112 and 114 positive) were excluded for analysis, interference between 110 and 112/114 was not an issue.

Comment 9:

Given that barcoding was done at the end of the staining procedures its purpose is to help eliminate doublets (this is not mentioned but should be added) and to run several samples simultaneously, with seemingly 17 total samples run not an absolute necessity. “To accommodate the required number of samples....Healthy donor was also included in batch as an internal control.” Please can the authors explain how the samples were run/batched, i.e. all samples from one individual plus HD in each case, or HD only once, or ?? How many

cells per sample were added into each batch? How did the authors handle cell fragility observed with longer exposure of cells suspended in water for injection? How did the authors control for potential batch effects (if applicable, but unclear if this is a possibility the way this is written).

Response:

We have now included the reasoning for the barcoding under the **Mass cytometry sample barcoding** section of the Supplementary Methods. We have also added a more detailed description of how the samples were batched and batch variation analysed in the **Mass cytometry sample batching and acquisition** section. We also discuss the technique of running batched samples in small aliquots to minimize the time spent in water.

Comment 10:

Please clarify if the healthy donor control was stained in the same way as the patient samples.

Response:

We have now clarified that the healthy donor control was prepared and stained in parallel with the volunteer samples in the section **Mass cytometry sample batching and acquisition**.

Comment 11:

Please specify for the mass cytometry runs, what number of total cells were acquired and what the total number of live MAIT cells in each measured sample has been given that their frequency is rather low. This will help understand the statistics on page 9 of the results section.

Response:

This information has now been added to the **Mass cytometry sample batching and acquisition** section and the **Mass cytometry data analysis** section of the Supplementary Methods.

Comment 12:

Mass cytometry data analysis: This section only roughly describes how normalization and de-barcoding was done but not how the actual mass cytometry data were analyzed. Please can the authors add this to the methods section.

Response:

We have now included information regarding how the MAIT cells were identified and the cloud based platform used for visualization and generation of results in the **Mass cytometry data analysis** section of the Supplementary Methods.

Comment 13:

Results from line 175 on page 9: The mass cytometry data confirm the observation already made by flow cytometry that CD38 is upregulated on MAIT cells in patients with enteric fever. The data further show an upregulation of Ki67 while none of the remaining markers from the 42 antigen panel showed any significant differences. As this is the only significant mass cytometry result the authors might choose to show the raw data plot of Ki67 vs CD38 at least exemplary for one patient in addition to the statistical workup. As n=4 is a rather low number a follow up on more individuals by regular flow cytometry focusing on CD38/Ki67 would be desirable. Previous publications [e.g Blood (2013), 121(7):1124 or Front Immunol (2017), 8:398] show a correlation of CD38^{hi}/Ki67 positive MAIT cells in HIV as well as after bacterial infection indicating a common trend in marker expression after immune challenge. These papers should be referenced. If the authors think the upregulation of Ki67 as not significantly relevant to the overall findings of the paper they may consider omitting the mass cytometry data altogether.

Response:

We agree that the Ki67 result from the CyTOF should be validated by flow cytometry. We have now added an additional supplementary figure to show the expression of Ki67 and CD38 on MAIT cells at Day 0, Day 5–10 and Day 28 in six additional volunteers who were diagnosis with enteric fever (**Supplementary Fig. 4**). This demonstrates that the Ki67⁺ MAIT cells are predominantly also CD38⁺ and only occur at the peak of infection (Day 5–10) compared to CD38, in which expression is maintained up to Day 28.

We have also now added the suggested references (**lines 176–7**).

Comment 14:

The authors keep changing the day numbering throughout the paper (e.g in Fig S2a it is 0/4/PD+96/28, in Fig 1 it is 0, 4, 7-8, 9-11, 28...), which is rather confusing and should be standardized for easier reading and comparison of the same type of cell samples.

Response:

We agree this is an inconsistency and have now changed all figures to the numbering style of Figure 1.

Comment 15:

Figure S2 a and b: Please clarify: in the results section the authors state that cells from 4 infected individuals were analyzed by mass cytometry. Can the authors please clarify what D7/18/19/20 stand for and if all samples were the PBMCs harvested at day 0/4/PD+96 and 28?

Response:

We agree this could be made clearer and have now spelled out the D as donor and have added a sentence in the figure legend regarding the PBMC timepoints that were harvested.

Comment 16:

The data from the healthy donor control are not shown for comparison in figure S2b, can the authors please include this as a baseline sample or state how these results were used for this manuscript.

Response:

The healthy donor was used only for assessing batch-to-batch variation, which we have now specified in the **Mass cytometry sample batching and acquisition** section of the Supplementary Methods.

Reviewer 3

We wish to thank this Reviewer for acknowledging that *“this is a remarkable study..... which offers a level of stringency normally only possible with mouse based experiments”*. *“This study has provided valuable insight into the influence of their TCRb chain on their expansion/contraction during the course of disease and its impact on their reactivity to antigen in general”*

Comment 1

My main concern with the study is that it relies on a surrogate phenotype for MAIT cells (V α 7.2 vs CD161 on CD4- T cells). While it is generally accepted that this is a fairly reliable way to identify MAIT cells in most healthy individuals, it is certainly possible that some

MAIT cells might downregulate CD161 (as reported during HIV infection) and other non-MAIT cells that are V α 7.2+ might upregulate CD161. This could easily confound the interpretation of the data in this paper because some MAIT cells might be escaping from the analysis gate while other non-MAIT cells might be moving in to that gate. This would explain why the typical MAIT TCR TRBV genes (6 and 20) are being replaced by other TRBV genes. It was therefore very reassuring at the end of the paper to see the data with Jurkat cells transduced with one representative TCR from the expanded and contracted populations, where they were clearly MR1 dependent. However, only one TCR being tested does not really negate the concern that many or most of these cells that become increased after infection might not be MAIT cells. The best way to validate the findings here would be to use MR1-5ARU/MG tetramers to investigate at least some of the key samples, to ensure that the MAIT cells are all still within the gates used, and that there is not an influx of non-MAIT cells in the same gates. The tetramers are available from NIH tetramer facility. If that is not possible, then at least analysis of a larger sample of the cells after infection for MR1 reactivity would be reassuring. For example, the cells could be sorted as a population and cocultured with the CD1-MR1 cells plus antigen and reactivity checked by ICS at the level of individual cells, and blocked with anti-MR1.

Response:

This is an important point and we agree that the gating strategy we used requires confirmation that the cells are MR1-restricted. To address this, we utilized MR1 tetramers and stained samples pre-, mid- and post- infection from Donor 4 and Donor 6 (**Supplementary Fig. 2**), as well as Donor 3 and Donor 5 (data not shown). We confirmed that the surrogate gated V α 7.2⁺ CD161⁺ MAIT cell were $\geq 96\%$ 5-OP-RU MR1 tetramer positive and thus, MR1-restricted and we did not observe any downregulation of CD161.

Comment 2

It is clear that this group has the ability to generate TCR transduced cell lines and this section (Figure 5) provided compelling data on the impact of two different TCR β chains, but unfortunately only one expanded and one contracted TCR β chain was compared. It isn't really possible to derive general conclusions from n=1 sample and this would be much better if 2-3 or more of each TCR β type (expanded vs contracted) was compared.

Response:

We agree that the addition of more expanded/contracted pairs would strengthen **Figure 5** and have therefore added in a second pair from Donor 3 to further support the conclusions drawn from Figure 5.

Comment 3

The focus on the TCR changes is on TRBV genes, but human MAIT cells can also vary for TRAJ genes (and possibly also TRAV genes although they would be excluded using the V α 7.2 gating approach). It would be useful to determine if the TCR α chain is also modulated during the course of infection.

Response:

We agree that, although the TCR α chain is restricted to TRAV1-2 in MAIT cells, changes could be occurring in the TRAJ usage in response to infection. To address this, we analyzed the TRAJ usage of MAIT cells in Donor 1 before, during and after infection shown in **Supplementary Fig. 9** and found the expected strict TRAJ usage with the predominant TRAJ33 was the most used with a small proportion using TRAJ12 and TRAJ20. The usage did not appear to change significantly in response to infection.

Comment 4

It is not appropriate to say there was a difference that was not statistically significant (p9). The null hypothesis that the groups are the same can't be rejected. However, it looks like the wrong type of ANOVA was used to determine this value. It is appropriate to use a 'repeated measures ANOVA' which would link samples from the same individual. If this was not used it should be because it will give more power to resolve differences within individuals.

Response:

We agree and have now removed the sentence, which states the decrease is not statistically significant (**lines 160–2**). We also agree that ideally a repeated measures ANOVA should be used to compare the changes occurring in these individuals over time, however, as not all individuals have measures in each timepoint unfortunately it is not possible to use repeated measures ANOVA for this figure. We have included the P-value for the undiagnosed individuals as it was close to significance at $P = 0.06$ in order to provide greater clarity on this observed decrease.

Comment 5

The IL12 blocking experiments (p10) seem to be overstated. Only 2 donors were tested and only for Salmonella was there an apparent decrease – not for E. coli, and no significance was established in either case.

Response:

We agree and have now improved the cytokine blocking experiment in **Supplementary Fig. 5**. Please see Reviewer 1 Comment 14 for a detailed explanation.

Reviewers' comments:

Reviewer #1 (Remarks to the Author):

The authors have done a solid job of addressing the concerns raised at review.

I recommend the paper as being ready for publication.

Reviewer #2 (Remarks to the Author):

After reviewing the additional information added to the manuscript the authors have generated sufficient clarity around the mass cytometry aspect of the work presented. No further additions are required, from a technical perspective the paper can be accepted.

Reviewer #3 (Remarks to the Author):

The authors have carried out a large amount of additional work in response to reviewers' comments and the paper is much improved as a result. I have only a couple of minor additional comments:

1. The fact that dialysed FBS provided a weaker response than non-dialysed FBS for the cell lines used in figure 5 does not necessarily indicate that the response was from vit-B metabolites. There are many elements in FBS that may be removed by dialysis that could impair the response of the cells. This could be easily tested by comparing dialysed and non-dialysed FBS in the presence of defined antigen such as EC5 or PT5 or 5ARU/MG. Or, irrelevant control cells that respond to other types of antigens such as aGalCer stimulation should also be tested. If these also show impaired response then it is unreasonable to state that the impaired response is due to dialysis removal of vitamin metabolites.

2. The revised sentence about how to describe a not-significant result still seems to be arguing for a difference even though the difference is not significant. The statement "...with a decrease in proportion to 0.76 ± 0.06 occurring only on day 10 ($P = 0.06$)" has still rejected the null hypothesis. Given that statistics has been employed to test if there is a difference, this needs to be rewritten that "no significant differences were observed for in individuals not diagnosed with infection (Figure 1b)". If the authors really wish to highlight the d10 result they could add a comment to the effect that some results approached significance ($p = 0.06$).

Point-by-point Response to Reviewer 3's Comments

Reviewer 3

The authors have carried out a large amount of additional work in response to reviewers' comments and the paper is much improved as a result. I have only a couple of minor additional comments.

Comment 1:

The fact that dialysed FBS provided a weaker response than non-dialysed FBS for the cell lines used in figure 5 does not necessarily indicate that the response was from vit-B metabolites. There are many elements in FBS that may be removed by dialysis that could impair the response of the cells. This could be easily tested by comparing dialysed and non-dialysed FBS in the presence of defined antigen such as EC5 or PT5 or 5ARU/MG. Or, irrelevant control cells that respond to other types of antigens such as aGalCer stimulation should also be tested. If these also show impaired response then it is unreasonable to state that the impaired response is due to dialysis removal of vitamin metabolites.

Response:

To determine whether dialyzed FBS impairs T cell activation, we designed an experiment to address this comment using a *Salmonella* Paratyphi specific HLA class II-restricted CD4⁺ T cell clone with a known peptide specificity and activated it with its specific peptide in normal versus dialyzed FBS. The result is shown in panel (a) below, demonstrating that the level of activation by CD69 expression is comparable between the dialyzed and non-dialyzed FBS in the media, and no impairment of activation is observed. Compared to panel (b) showing the MR1-restricted MAIT.Jurkat activation in dialyzed versus non-dialyzed FBS without stimulation. This result provides compelling evidence in support of the conclusion that the decrease in CD69 expression by MAIT.Jurkat in dialyzed FBS is specific for MR1-restricted T cells.

Dialyzed FBS does not inhibit activation of peptide-specific T cells. (a) 100 000 CD4⁺ T cells specific for the *Salmonella* protein CdtB were stimulated with 3 µg/mL CdtB 105–125 peptide for 18 hours. They were then stained for flow cytometric analysis using anti-CD69 (A488) conjugated antibody. (b) Expression of CD69 on MR1-restricted Jurkat.MAIT line in the presence of normal versus dialyzed FBS as described in the revised manuscript.

We have now also modified our revised manuscript to refer to the lack of generalized activation impairment by dialyzed FBS (**line 278**).

Comment 2:

The revised sentence about how to describe a not-significant result still seems to be arguing for a difference even though the difference is not significant. The statement “...with a decrease in proportion to 0.76±0.06 occurring only on day 10 (P = 0.06)” has still rejected the null hypothesis. Given that statistics has been employed to test if there is a difference, this needs to be rewritten that “no significant differences were observed for in individuals not diagnosed with infection (Figure 1b)”. If the authors really wish to highlight the d10 result they could add a comment to the effect that some results approached significance (p = 0.06).

Response:

To address this comment, we have now changed the sentence in the revised manuscript to state that no significant differences occurred (**line 161**).

REVIEWERS' COMMENTS:

Reviewer #3 (Remarks to the Author):

The authors have done a commendable job at addressing all of my concerns and I have no additional comments.